# Plug-and-Play Bayesian Online Change-Point Detection in Gaussian Processes

## Abstract

We study the problem of detecting non-stationarity in online time series, when the underlying distribution is assumed to be a piecewise 1D-Gaussian process. Drawing inspiration from Bayesian online change-point detection methods such as that of Adams and MacKay [2007], we construct a restarted variant of that to specifically deal with arbitrary changes both in mean and variance of 1D-Gaussian processes. We evaluate our algorithm on both synthetic datasets of varying task difficulty and on prevalent real-world data across a variety of fields. Our results compare favorably with state-of-the-art, as measured by the detections' F1-SCORE. Code will be provided to ensure easy reproducibility.

## 1 Introduction

While most statistical models assume that the input data is generated according to some underlying distribution, it is often assumed that the latter does not actively change throughout instances of training or inference. This presents a major setback when tackling real-world problems where the data collection process is often done in a sequential manner and where the parameters of the generating distribution, if assumed parametric, are changing continuously. This applies in various ways across a variety of fields in machine learning and statistics: distribution shifts in deep learning (DL), environment non-stationarity in reinforcement learning (RL), to name a few.

More precisely, we consider the general setting of a sequential decision-making process, during which a series of abrupt changes, commonly referred to as *change-points* (CP), take place. Change-points are sudden shifts in the underlying parameters of the data generating distribution of a given sequence. Detecting these change-points in real-time is of vital importance for the analysis and forecasting of time series, especially in high volatility scenarios, in consumer decision modeling (Xu and Yun [2020]), service provider adaptation to customers, and pricing (Taylor [2018], Kanoria and Qian [2019], Bimpikis et al. [2019], Gurvich et al. [2019]), wireless communication networks (Zhou and Bambos [2015], Zhou et al. [2016]), epidemic networks and control (Nowzari et al. [2016], Kiss et al. [2017]), inventory management (Agrawal and Jia [2019], Huh and Rusmevichientong [2009]), non-stationary multi armed bandits (Alami et al. [2017], Alami and Azizi [2020], Alami [2023, 2018], Garivier and Moulines [2011], safety-aware bandits: Alami et al. [2023a]), RL [Alami et al., 2023b], and notions of automated quality control (El Mekkaoui et al. [2024]), to name a few.

**Key contributions.** We outline our contributions as follows

- We propose a novel variant of the Restarted Bayesian Online Change-Point Detection algorithm (R-BOCPD), that generalizes the modeling scope to the setting where online observation stream is generated from an underlying piecewise stationary 1D-Gaussian process (GP). Our model incorporates changes in either means or variances (or both) of the underlying GP.

- We demonstrate our results experimentally across a wide range of tasks of varying difficulty, on both synthetic and real-world datasets. Our algorithm compares favorably to

Submitted to Workshop on Bayesian Decision-making and Uncertainty, 38th Conference on Neural Information Processing Systems (BDU at NeurIPS 2024). Do not distribute.

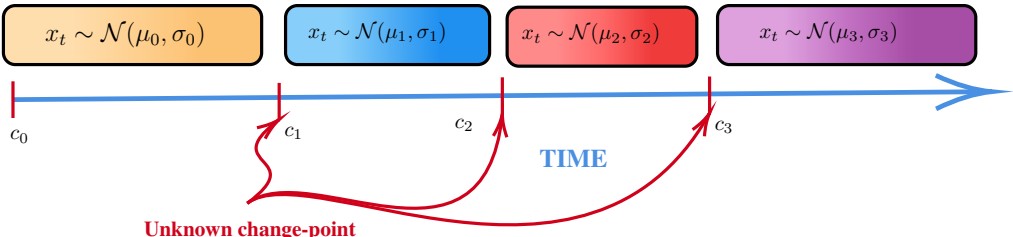

Figure 1: Piecewise Stationary Gaussian Process. Starting from change-point $c_\ell$, the model is assumed to be Gaussian of parameters $(\mu_\ell, \sigma_\ell)$.

state-of-the-art in both online and offline settings, as measured by false-alarm and misdetection rates, detection delay, and finite-time runtime.

- We present concrete promising directions for future work relevant both to theoreticians and practitioners working both on probabilistic modeling under uncertainty and general modeling of time-series data in a variety of fields.

## 2 Main Results

Given the prevalent use of 1D-Gaussian distributions as statistical priors for a wide range of problems across a variety of domains, we choose to model a piecewise stationary GP, where a set of unknown abrupt change-points $\{c_\ell\}_{\ell=1}^{L}$ take place such that

$$x_t \sim \mathcal{N}(\mu_\ell, \sigma_\ell) \quad \forall t \in [c_\ell, c_{\ell+1}], \forall \ell \in [1, L] \tag{1}$$

where $L$ is the unknown total number of change-points throughout $[1, t]$. We provide Figure 1 for visualization.

**Challenge.** We aim to detect the change-points online and sequentially from data in real-time, without *à-priori* knowledge on their number, their location, or how often they occur. The latter, indeed, can be used as a plug-and-play prior, as showcased later on in Algorithm 1. In particular, we are interested in designing an algorithm that detects change-points

- *reliably*, with as few misdetections and false-alarms as possible.

- *in real-time*, with as low of a detection delay as possible.

- *incorporating uncertainty*, yielding statistically optimal or near-optimal probabilistic guarantees, allowing for flexibility and control to the decision make.

### 2.1 Change-point Detection as *Runlength Inference*

We first introduce the notion of *runlength* $r_t$, which is defined as the number of time steps since the last change-point to the process, given an observed data sequence $\mathbf{x}_{1:t}$ (up to current time step $t$). Adams and MacKay [2007] introduce an efficient Bayesian approach for handling piecewise stationary processes via computing the posterior distribution over the current runlength $r_t$. The exact inference on the runlength distribution is done recursively via message-passing as follows

$$p\left(r_t|\mathbf{x}_{1:t}\right) \propto \sum_{r_{t-1}} \underbrace{p\left(r_t|r_{t-1}\right)}_{\text{hazard function}} \underbrace{p\left(x_t|r_{t-1}, \mathbf{x}_{1:t-1}\right)}_{\text{UPM}} p\left(r_{t-1}|\mathbf{x}_{1:t-1}\right)$$

where the *hazard function* is defined as

$$p\left(r_t|r_{t-1}\right) = \begin{cases} H\left(r_{t-1}\right) & \text{if } r_t = 0 \\ 1 - H\left(r_{t-1}\right) & \text{if } r_t = r_{t-1} + 1 \\ 0 & \text{otherwise} \end{cases} \tag{2}$$

where $H$ is defined as $H(s) = \frac{\mathbb{P}_{\text{change}}(s+1)}{\sum\limits_{t=s+1}^{\infty} \mathbb{P}_{\text{change}}(t)}$, $\mathbb{P}_{\text{change}}$ denotes the probability distribution over the interval between changepoints, and the *underlying probability model* (UPM) depends on the probability distribution of $x_t$. We showcase an illustrating example for our reasoning and algorithmic construction in Appendix A.

---
**Algorithm 1** R-BOCPD for Piecewise Stationary GPs
---
**Input:** $h \in (0,1), \alpha_0 = 1, \beta_0 = 1, n_0 = 1, \mu_0 = 0$

1:  $r \leftarrow 1, \nu_{r,r:r} \leftarrow 1, \mu_{r,r:r} \leftarrow \mu_0, \alpha_{r,r:r} \leftarrow \alpha_0, \beta_{r,r:r} \leftarrow \beta_0, n_{r,r:r} \leftarrow n_0$.

2: **for** $t = 1, \ldots$ **do**

3:     **Observe** $x_t \sim \mathcal{N}(\mu_t, \nu_t)$

4:     **For each forecaster starting at time $r$ and $s$ up to time $t$, define**

$$\nu_{r,s:t} \leftarrow \begin{cases} (1-h)\exp\left(-\ell_{s:t}\right)\nu_{r,s:t-1} & \text{for } s \in [r,t), \\ h \sum\limits_{i=r}^{t-1} \exp\left(-\ell_{i:t}\right)\nu_{r,i:t-1} & \text{for } s = t \end{cases} \tag{3}$$

5:     **Estimate the last change-point** $\widehat{\tau}_t$:    $\widehat{\tau}_t \leftarrow \mathrm{argmax}_{s \in [r,t]} v_{r,s:t}$

6:     **if** $\widehat{\tau}_t = t$ **then**

7:        $r \leftarrow t + 1, \nu_{r,r:r} \leftarrow 1, \mu_{r,r:r} \leftarrow \mu_0, \alpha_{r,r:r} \leftarrow \alpha_0, \beta_{r,r:r} \leftarrow \beta_0, n_{r,r:r} \leftarrow n_0$.

8:     **end if**

9: **end for**

---

## 2.2 Algorithmic Construction of R-BOCPD

For an algorithmic construction that adheres to our previously outlined design objectives, we draw inspiration from Alami et al. [2020b]'s extension to the seminal *Bayesian Online Change-point Detection* (BOCPD) work of [Fearnhead and Liu, 2007]. Indeed, Alami et al. [2020a] introduce a pruned variant of the latter, which they refer to as the *Restarted Bayesian Online Change-point Detection*, particularly designed to model online changes in the means of univariate Bernoulli-distributed data samples. We propose to extend the R-BOCPD construction to change-points in an online piecewise stationary GP. In particular, we start by introducing a few definitions in the following

**Definition 2.1** (Predictor). Given a sequence of observations $\mathbf{x}_{s:t} = (x_s, \ldots, x_t)$, we define an instance of a predictor as follows

$$\mathrm{PRED}\left(x_{t+1}|\mathbf{x}_{s:t}\right) = \frac{\Gamma\left(\frac{2\alpha_{s:t}+1}{2}\right)}{\Gamma\left(\alpha_{s:t}\right)\sqrt{2\alpha_{s:t}\pi}}\left(1 + \frac{1}{2} \times \frac{(x_{t+1} - \mu_{s:t})^2}{\frac{\beta_{s:t} \times (n_{s:t}+1)}{n_{s:t}}}\right)^{-\frac{2\alpha_{s:t}+1}{2}} \tag{4}$$

where $\mathrm{PRED}\left(x|\emptyset\right) = \frac{\Gamma\left(\frac{2\alpha_0+1}{2}\right)}{\Gamma(\alpha_0)\sqrt{2\alpha_0\pi}}$ for some chosen $\alpha_0 > 0$ at initialization and incrementing procedure

$$\alpha_{s:t+1} = \alpha_{s:t} + \frac{1}{2} \quad n_{s:t+1} = n_{s:t} + 1 \quad \beta_{s:t+1} = \beta_{s:t} + \frac{n_{s:t} \times (x_{t+1} - \mu_{s:t})^2}{2 \times (n_{s:t}+1)}$$

Akin to the our considered construction, instead of dealing with run-length, we introduce the notion of forecaster loss as loss incurred by predictors across time

**Definition 2.2** (Forecaster Loss). Using the predictor, the instantaneous loss of the forecaster $s$ at time $t$ is given by:

$$\ell_{s:t} := -\log \mathrm{PRED}\left(x_t|\mathbf{x}_{s:t-1}\right).$$

Then, let $\widehat{L}_{s:t} := \sum\limits_{s'=s}^{t} \ell_{s':t}$ denotes the cumulative loss incurred by the forecaster $s$ from time $s$ until time $t$ which takes the following form

$$\widehat{L}_{s:t} := \sum_{s'=s}^{t} -\log \mathrm{PRED}\left(x_t|\mathbf{x}_{s':t-1}\right) \tag{5}$$

Thus, the forecaster weights update will remain the same (for some parameter $h \in (0,1)$).

$$\nu_{r,s:t} = \begin{cases} (1-h)\exp\left(-\ell_{s,t}\right)\nu_{r,s:t-1} & \forall s < t, \\ h \times V_{r:t-1} & s = t. \end{cases} \quad \text{with} \quad V_{r:t-1} := \sum_{i=r}^{t-1} \exp\left(-\ell_{i:t}\right)\nu_{r,i:t-1} \tag{6}$$

Finally, we keep the same restart procedure as in Alami et al. [2020b], namely

$$\mathrm{RESTART}(x_r, \ldots, x_t) = \mathbb{I}\left[\exists\, s \in (r,t) : \nu_{r,s:t} > \nu_{r,r:t}\right] \tag{7}$$

where $\mathbb{I}(.)$ is the indicator function. Finally, we describe our algorithm in full pseudo-code in Algorithm 1.

| ALGORITHM | DATASET | | | | | | | |
|---|---|---|---|---|---|---|---|---|
| | VERY EASY | | EASY | | MEDIUM | | HARD | |
| | F1-SCORE ↑ | DELAY ↓ | F1-SCORE ↑ | DELAY ↓ | F1-SCORE ↑ | DELAY ↓ | F1-SCORE ↑ | DELAY ↓ |
| BINSEG [Scott and Knott, 1974] | 0.87 | — | **1.0** | — | 0.8 | — | 0.6 | — |
| PELT [Killick et al., 2012] | **1.0** | — | **1.0** | — | 0.909 | — | 0.67 | — |
| CUSUM[1][Basseville and Nikiforov, 1993] | 0.632 | 0.444 | — | — | — | — | — | — |
| GLRT [Keshavarz et al., 2018] | **1.0** | 0.009 | **1.0** | 0.14 | 0.889 | 0.339 | 0.174 | 0.879 |
| R-BOCPD (OURS) | **1.0** | 0.01 | **1.0** | 0.016 | **1.0** | 0.038 | **0.9** | **0.498** |

Table 1: Benchmark results for various algorithms across different synthetic datasets of varying difficulty. We report the F1-SCORE and DELAY for each method. Delay is normalized by the size of the stationary periods for each detected change-point for appropriate unified performance assessment across sequences of different lengths. We highlight here that the first family of methods is offline, hence naturally would get much smaller delays. The second group of methods is online, akin to our proposed method. For the interest of comparison within our considered setting, we only measure detection delays for online methods.

## 2.3 Empirical Results

**Measuring Performance.** We design a diverse synthetic task suite and also evaluate on prevalent open-source real-world time-series data (https://github.com/alan-turing-institute/TCPD). We evaluate our algorithm on two key metrics: *F1-Score* and *detection delay*. We normalize the latter by the length of its corresponding stationary period, i.e for change-point $c_\ell$, delay $d_\ell$ is normalized by $c_{\ell+1} - c_\ell$.

| ALGORITHM | DATASET | | | |
|---|---|---|---|---|
| | JFK PASSENGERS | CO₂ CANADA | BUSINV | BITCOIN |
| BINSEG [Scott and Knott, 1974] | 1.0 | 0.67 | 0.24 | 0.43 |
| PELT [Killick et al., 2012] | 0.5 | 0.67 | 0.20 | 0.43 |
| GLRT [Keshavarz et al., 2018] | **1.0** | 0.8 | 0.57 | 0.58 |
| RBOCPDMS [Knoblauch et al., 2018] | — | — | 0.27 | — |
| GPTS-CP [Saatçi et al., 2010] | — | — | 0.62 | — |
| ADAGA [Caldarelli et al., 2022] | — | — | 0.77 | — |
| R-BOCPD (OURS) | **1.0** | **1.0** | **0.8** | **0.8** |

Table 2: Benchmark results for various algorithms across different real-world datasets belonging to a variety of domains. We mainly compare in terms of F1-SCORE. We were unable to reproduce RBOCPDMS, GPTS-CP and ADAGA at the time of submission. Instead, we reproduce their exact setting for the BUSINV dataset.

**Runtime.** Given the online nature of our proposed algorithm, we analyze to what extent our algorithm (and others) allow for smooth inference in real-time. We see that our runtime compares favorably with state-of-the-art, which makes it especially useful for modeling long-context sequences.

| ALGORITHM | RUNTIME (MS) ↓ |
|---|---|
| BINSEG [Scott and Knott, 1974] | 2.82666 |
| PELT [Killick et al., 2012] | 15.3721 |
| GLRT [Keshavarz et al., 2018] | 3.0112 |
| R-BOCPD (OURS) | **0.7097** |

Table 3: Runtime (in milliseconds) per iteration. The first family of methods is offline, while the second one is online, similarly to R-BOCPD.

**Discussion & Future Work.** Our work sheds light on online change-point detection when the underlying distribution of the streaming data is a 1D-Gaussian. Being both quite relevant in practice and theoretically interpretable, this lays the ground for a variety of possible extensions and future work. Among these we list- establishing potential theoretical optimality guarantees, scaling to mixtures of detectors and building hybrid algorithms. We defer a detailed discussion of the latter to Appendix B.

---

[1]CUSUM primarily detects shifts in the mean of a process and does not account for changes in the covariance structure. To fairly showcase how it performs, we design the "Very Easy" setting where only changes in the mean of underlying process occur.

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

# Supplementary Material

## Table of Contents

## A Learning through "Runlength" Inference

A simple example of the aforementioned approach would be to use a constant hazard function $h$ taking some value in $(0, 1)$. Given that, In the sense that $p(r_t = 0|r_{t-1})$ is independent of $r_{t-1}$ and is constant, giving rise, à priori, to geometric inter-arrival times for change points ($\mathbb{P}_{\text{change}}(s + 1) = h(1 - h)^s$). Thus, the recursive runlength distribution computation becomes:

$$p(r_t \neq 0|\mathbf{x}_{1:t}) \propto (1 - h)\, p(x_t|r_{t-1}, \mathbf{x}_{1:t-1})p(r_{t-1}|\mathbf{x}_{1:t-1})$$

$$p(r_t = 0|\mathbf{x}_{1:t}) \propto h \sum_{r_{t-1}} p(x_t|r_{t-1}, \mathbf{x}_{1:t-1})p(r_{t-1}|\mathbf{x}_{1:t-1})$$

## B Future Work

We list a few directions for future work, as outlined in the discussion.

- Establishing potential theoretical optimality guarantees (for instance in terms of detection delay and false-alarm rate, akin to that in Alami et al. [2023b] in the case where the underlying distribution is piecewise multinomial), given the particular algorithm construction we adopt.

- Scaling to online probabilistic (posterior-weighted) mixtures of forecasters, which would reduce the sensitivity to the prior choice of $h$ in Algorithm 1 and would potentially allow to systematically increase the confidence predictions given an increase in resources.

- Given the increasing availability of ground-truth annotators such as that of van den Burg and Williams [2022], this would allow to potentially build hybrid algorithms on top where parts of the algorithm can be learned/tuned from past data.

