# OpenReview forum: "Plug-and-Play Bayesian Online Change-Point Detection in Gaussian Processes"
_NeurIPS.cc/2024/Workshop/BDU — Submitted to NeurIPS BDU Workshop 2024_

### Official Review · Reviewer_rCpc · 2024-09-23
**review comments for Submission 106**

**Rating:** 7
**Confidence:** 3

**Review:**

The paper proposed the restarted Bayesian online change-point detection. The paper is well-organized. I have a few comments about the result delivery:

1. It would be helpful to more explicitly highlight how this work improves upon or differs from previous approaches, especially in comparison with other GP-based models for change-point detection.
2. The experimental validation appears robust. However, it would be insightful to expand the discussion on the limitations or edge cases where the model may not perform as well.
3. it would be beneficial to discuss the scalability issue, especially for real-time large-scale applications

---

### Official Review · Reviewer_w8Wd · 2024-09-24
**Science requires clear and complete documentation**

**Rating:** 3
**Confidence:** 4

**Review:**

This must not be accepted. The paper is grossly incomplete and poorly organized.

Pros: It looks like it might, potentially, give a useful extension to the R-BOCPD framework. The suite of empirical tests is compelling, albeit lacking.

Regarding completeness:
A reader cannot gather even the basic premise of what is happening in Section 2.2 without at least consulting Alami et al. 2020. The authors do not at all justify the claim that this method is "theoretically interpretable," as stated on line 101. It is incumbent on the authors to explain exactly what they are doing, why they are doing it that way, and how they accomplish all of their stated aims. They have failed to adequately do so, which is the major reason why I say the paper is incomplete. In particular, it is not sufficient to merely gesture at previously-published literature, with the vague sense that it might offer a more complete account of perhaps some important points.

The superior empirical test results are the key demonstration the authors give for why anyone should care about this method, so adequately describing the scientific basis of the tests is essential, which includes specifying the dataset composition as well as the metrics used. The authors do not state how the datasets referenced in Table 1 were constructed, aside from the footnote stating that the Very Easy dataset has "only changes in the mean of the underlying process"; at most, one can infer that the other datasets include changes in both the mean and variance. The key "Delay" metric is broadly described but not precisely defined. These omissions make it harder to judge the significance of the reported results. The authors do not show the performance of traditional BOCPD, which would go some way toward demonstrating what value this new method adds (if such a comparison is not relevant, this should be directly addressed, because it's the obvious and glaring question).

The authors go out of their way to emphasize that they are interested in "incorporating uncertainty, yielding statistically optimal or near-optimal probabilistic guarantees" (lines 56-57). Nowhere do they demonstrate or even discuss how their new method addresses this objective. In comparing the runtimes in Table 3, the authors do not address the question of whether the results are due more to differences in the underlying algorithms, or instead to differences in the code implementations, which may be more or less highly optimized. The abstract promises that "code will be provided to ensure easy reproducibility" (line 10). If the code exists now in sharable form, it should be linked in the paper. If it does not, then it is misleading to advertise this in the abstract; it could instead be mentioned under "Future Work."

Regarding organization:
I am unsure what Section 2.1 adds to the discussion. It is presented as if it provides illuminating background for Section 2.2, but it is not clear how. Appendix A is unnecessary, given that the rest of the paper makes no use of it, and that the primary goal of this paper is not an explanation of the old method but instead the new method of Section 2.2. Appendix B could very naturally have been included in the main body of the text, and it would have saved some space to write it as a standard paragraph instead of as a bulleted list. These aspirations for future work certainly do not constitute a "key contribution" of the present paper, as asserted in lines 41-43.

Equation 5 is an utterly trivial consequence of Definition 2.2 combined with the definition of L_s:t. It does not need to be stated and the space could be better used for something else. For example, making the font in the tables larger. The text of Table 1 in particular is unacceptably small. I would suggest splitting the F1 score and Delay comparisons into separate tables, to avoid the large amount of wasted space taken up by the BINSEG and PELT methods under the Delay columns. If the authors are already reporting the Delay comparison, then a separate Runtime comparison (Table 3) has relatively low value.

Other notes:
Alami et al.'s "Restarted Bayesian online change-point detector achieves optimal detection delay" is repeated twice in the list of References (lines 124-126, and again on lines 127-131). As a result, it is referred to as both Alami et al. [2020a] and [2020b] in the text. This is especially egregious because it is obligatory to actually seek out and read through this reference in order to understand what's going on here.
van den Burg and Williams [2022] should be cited at the introduction of TCPD (line 94).

---

### Decision · Program_Chairs · 2024-10-09

**Decision:**

Reject

**Comment:**

Reviews for this paper are mixed. The high scoring review is very short, while the low-scoring review is very details and complains mainly that the work seems unfinished and needs additional polish. While submitting workshops based on work-in-progress is encouraged, my take from reading the review is that this particular paper is not quite complete enough to be ready to present. As consequence, I encourage the authors to continue this work, polish it up, and resubmit to a future workshop.